# Reinforced Epoxy Binder Modified with Borpolymer

**DOI:** 10.3390/polym15122632

**Published:** 2023-06-09

**Authors:** Aleksei G. Tuisov, Aisen Kychkin, Anatoly K. Kychkin, Elena S. Anan’eva

**Affiliations:** 1Federal Research Center, The Yakut Scientific Centre of the Siberian Branch of the Russian Academy of Sciences, Yakutsk 677000, Russia; 2V.P. Larionov Institute of Physical and Technical Problems of the North Siberian Branch Russian Academy of Sciences, Yakutsk 677980, Russia; kychkinplasma@mail.ru; 3Department of Modern Special Materials, Polzunov Altai State Technical University, Barnaul 656038, Russia; 4Department of Nanocomposite Materials, Novosibirsk State University, Novosibirsk 630090, Russia

**Keywords:** epoxy resin, boron polymer, viscosity, solubility, tensile strength at tension tensile strength, physical and mechanical properties

## Abstract

Polymer binders based on epoxy resins have unique properties that contribute to their use in many composite industries. The potential of using epoxy binders is due to their high elasticity and strength characteristics, thermal and chemical resistance, and resistance to climatic aging. This is the reason for the existing practical interest in modifying the composition of epoxy binders and understanding the strengthening mechanisms in order to form reinforced composite materials with a required set of properties based on them. This article presents the results of a study of the process of dissolving the modifying additive of polymethylene-p-triphenyl ether of boric acid in the components of an epoxyanhydride binder applicable to the production of fibrous composite materials. The temperature and time conditions for the dissolution of polymethylene-p-triphenyl ether of boric acid in anhydride-type isomethyltetrahydrophthalic anhydride hardeners are presented. It has been established that the complete dissolution of the borpolymer-modifying additive in iso-MTHPA occurs at a temperature of 55 ± 2 °C for 20 h. The effect of the modifying additive of polymethylene-p-triphenyl ether of boric acid on the strength properties and structure of the epoxyanhydride binder has been studied. Increases in transverse bending strength up to 190 MPa, elastic modulus up to 3200 MPa, tensile strength up to 0.8 MPa, and impact strength (Charpy) up to 5.1 kJ/m^2^ are observed when the content of the borpolymer-modifying additive in the composition of the epoxy binder is 0.50 mass. %.

## 1. Introduction

Current practice in the production of structural composite materials shows that the most accessible and well-recommended binders are currently epoxy binders and heat-curing compounds [1,2,3]. Their widespread use in engineering is associated, firstly, with the high manufacturability of epoxy resins, and secondly, with a unique combination of performance characteristics of products for their curing. Epoxy resins are oligomers that contain epoxy groups and are capable of forming crosslinked polymers under the influence of curing agents (such as anhydrides, amines, polyamines, etc.). Epoxy resins based on the polycondensation of epichlorohydrin with phenols, typically bisphenol A, have gained the most widespread use. The high reactivity of the epoxy group and the thermodynamic compatibility of epoxy oligomers with many substances make it possible to use a variety of hardeners and carry out curing reactions under various technological conditions. Features of the synthesis processes such as the absence of volatile products and low shrinkage are equally important.

Epoxy polymers have high values of static and impact strength, hardness and wear resistance. They are characterized by significant temperature and heat resistance. Many hard surfaces form strong adhesive bonds with epoxy polymers, which explains their use as adhesives, paints, coatings, and binders in composites.

In the manufacture of fiberglass plastics, in particular, three-component epoxy resin with dicarbonic acid anhydrides as a hardening agent is widely used. These epoxy–anhydride resins obtain high-strength properties, low settling, and high adhesion to fibrous fillers. However, epoxy–anhydride resins have some drawbacks, e.g., low heat resistance (within 120 °C), resistance to aggressive environments, flammability, etc.

Further extension of the field of application of epoxy–anhydride resins is possible by means of chemical and physico-chemical modification, changing properties of the epoxy binder [4,5,6,7,8,9]. Chemical modification involves altering the polymer network structure by adding compounds that become part of the polymer composition. For example, the addition of peroxides (simple polyether alcohols containing glycidyl groups, such as glycerin anhydride) provides elasticity to the cured resin by increasing the molecular weight of the interstitial fragment but reduces its water resistance. Adding halogen and phosphorus organic compounds increases the flame retardancy of the resin. Incorporating phenol-formaldehyde resins allows the epoxy resin to cure through direct heating without a curing agent, providing greater stiffness and improved anti-friction properties, but reducing impact viscosity. Physical modification is achieved by adding substances to the resin that do not form a chemical bond with the binder. For example, adding rubber increases the impact viscosity of the cured resin. The addition of colloidal titanium dioxide increases its refractive index and imparts opacity to ultraviolet radiation. The search for chemical substances which allow us to modify the structure and, correspondingly, the properties of a resin is an important research and implementation objective.

It is known that boron compounds [10], in particular, polyesters and boric acid polymethylene esters, are strongly adhesive to glass, metals, and wood and are now mainly used as adhesive compositions, plasticizers, lubricants, emulsifiers, etc. The boron compound boric acid polymethylene-p-triphenyl ether synthesized by the authors of [11,12,13,14,15,16,17,18] in their work has interesting and unique properties.

Thus, a relevant research objective is to assess the effectiveness of the application of boric acid polymethylene-p-triphenyl ether for modification of the structure and properties of epoxy–anhydride resins and the development of a combination technology through the solubility phase of boric acid polymethylene-p-triphenyl ether additives in components.

The innovativeness of the proposed solution lies in the use of a newly synthesized compound that is not currently used by any research group involved in the modification of epoxy binders.

## 2. Materials and Methods

A polymer binder based on epoxydine resin, isomethyl tetrahydrophalic anhydride and amine hardening reaction accelerator was selected as the subject of this study.

Epoxy resin is a soluble and fusible reactive oligomeric product based on epichlorohydrin and diphenylolpropane. The designation of the resin consists of the following: E-epoxy; D-diphenylolpropane; digits indicating the limit of the norm for the content of epoxy groups. Isomethyl anhydrite (iso-MTHPA) is a liquid mixture of isomers of methyltetrahydrophthalic anhydride. Crystallization is allowed when stored below 20 °C. The curing accelerator is an individual substance: 2,4,6-tris (dimethylaminomethyl) phenol (Agidol-53). Comparative characteristics of the binder components are shown in Table 1.

The modifier used was borpolymer powder produced by LLC “Boroplast”, whose structural formula is shown in Figure 1.

Table 2 shows the basic physico-chemical properties of the modifying additive boron polymer.

During microscopic examination using the electron microscope JEOL JSM 7800F, morphology and dispersion of borpolymer powder, shown in Figure 2, were researched. In its primary form, borpolymer is a polydisperse powder with maximum inclusion sizes of up to 150 µm.

Primarily, a study on the impact of a boron polymer on technological performance was of practical interest, and the following were selected as the characteristic parameters: time of “life”, time of gel formation and viscosity. Of secondary interest was the evaluation of the effectiveness of the use of a boron polymer as a modifier for changing the elastic performance (comparative tests) of modified resin regarding benchmarks. For this purpose, a test program was formed and implemented (Table 3).

### 2.1. Paint and Lacquer Materials: Method for Determination of Relative Viscosity

When thermally active polymers are exposed to heat, their viscosity begins to increase intensively, which marks the beginning of the process of gel formation. The use of a binder for the manufacture of articles or for impregnation is required in the period from the time of preparation of the binder to the moment when it becomes gel-like. This time span is called the lifetime (the viability of the binders).

Under normal conditions, the process of gel formation is long enough, so experiments determine the time before the start of gel formation at elevated temperatures and then recalculate the process to specified external conditions.

The initial conditional viscosity of the epoxy composition was determined on the capillary viscometer VZ-1, which is used for measuring the viscosity of unstructured and loosely structured liquids. The diagram of the VZ-1 nozzle is shown in Figure 3. The diameter of the viscometer nozzle is 5.4 mm.

The gel-forming time of the binder was determined by heating the epoxy binder to a temperature of 120 ± 2 °C on a tile with a hole diameter of 20 mm and a depth of 5 mm.

The epoxy binder is applied to the central part of the plate in a volume of 1.57 cm^3^ to determine the time of gel formation. The time, in seconds, between the application of the epoxy binder to the plate and the moment of the breakage of the strands is taken as the time of gel formation. The result of the test shall be the arithmetic mean of the three parallel definitions, the difference between the most different values not exceeding 5 s. The allowed relative total error of the test result shall be 3 percent with a probability of 0.95.

### 2.2. Method of Static Bending Test

During the three-point bending test, the test sample of a rectangular cross section lies freely on supports and is subjected to bending at a constant speed between the supports until the sample breaks or until the sample has reached a given value of relative deformation or bending. During the test, the load applied to the sample and the corresponding deformation halfway between the supports shall be measured. The diagram of the test is shown in Figure 4.

### 2.3. Plastics: Tensile Test Method

Figure 5 shows the dimensions of the tensile test samples according to GOST 11262-80. This form of specimen is applicable for isotropic materials, in particular for cured epoxy compositions.

### 2.4. Plastics: Method for Determination of Charpy’s Impact Strength

The determination of the Charpy impact viscosity for rectangular cut samples without incision (Figure 6) allows us to estimate the initiation energy of structural damage in the form of cracks in samples of a cured epoxy binder. Comparison of results before and after modification is an indicator of boron polymer efficiency as a component that increases material crack resistance.

### 2.5. Structure Study by SEM and IR Spectrometry

The structure of the cured samples of the epoxy–anhydride binder was studied using scanning electron microscopy methods based on the JEOL high-resolution JSM-7800F and IR spectrometry of liquid samples of the binder and modifying components on the instrument “Vecnor-22” in tablets with KBr.

An uncured binder (item 2.1) and samples of a cured binder without a modifying additive (items 2.2–2.5) were used as reference samples. The test results of the reference samples were compared with the results of samples containing the additive. Based on the results of the comparison, the effectiveness of the modification and the optimal degree of filling were evaluated.

## 3. Results

### 3.1. Results of a Study on the Decomposition of the Boron Polymer Additive in the Components of the Epoxyanhydride Binder

The solubility of solids in liquid solvents is highly influenced by the temperature at which the dissolution process takes place. In general, higher temperatures lead to increased solubility of most solids. The temperature at which the solid phase transitions into solution and the duration of the solubility process are the key factors affecting solubility. For the modifying additive, the following components of the epoxy–anhydride binder were chosen as potential solvents: iso-MTHPA (isomethyltetrahydrophthalic anhydride) hardener comprising 40–60% of the total binder mass, ED-22 epoxy resin (with epoxy group mass fraction of 22.1–23.6%) comprising 40–60% of the total binder mass, and 2,4,6-tri-N, N″-di-methylaminomethylphenol (Agidol-53) polymerization reaction accelerator, which constitutes 0.2–2% of the total binder mass.

The dissolution process of the boron polymer in individual components of the epoxy binder was conducted at a temperature of 50 ± 2 °C for 20 days, with a modifying additive content ranging from 1% to 6% by mass of the possible solvent. The degree of modifier dissolution was assessed by comparing the initial conditional viscosity of the solution with different contents of the modifying additive boron polymer to a control sample without the modifier. Table 4 presents the initial conditional viscosity values, determined at 40 °C, for solutions of boron polymer + ED22 and boron polymer + Agidol-53, aged at 50 ± 2 °C for 20 days.

Analysis of the results obtained shows that as the content of the modifying additive boron polymer + ED-22 mixture increases, the initial conditional viscosity of the boron polymer + ED-22 mixture does not change; therefore, the modifying additive boron polymer does not dissolve in epoxy resin. An assessment of the dilution of the boron polymer in the amine accelerator environment, based on a comparison of the initial conditional viscosity values, showed that as the boron polymer content in the Agidol-53 polymerization reaction accelerator increased, the initial conditional viscosity of the system does not change. That is, when you add the modifying additive boron polymer to Agidol-53 in the amount of 1 to 6 masses, no dilution process is observed, as shown by the initial conditional viscosity values (at 40 °C) of the boron polymer + Agidol-53 presented in Table 4. Figure 7 illustrates the relationship between the content of the modifying additive iso-MTHPA boron polymer and the initial conditional viscosity of the mixture at a temperature of 40 °C, sustained for 20 days at a temperature of 50 ± 2 °C.

Figure 7 illustrates the change in the initial conditional viscosity of the boron polymer + iso-MTGFA solution. Under the experimental conditions, the maximum boron polymer filling degree was 5 wt. %. The figure indicates that the pure iso-MTHPA hardener has an initial conditional viscosity of 8 s. As the content of the modifying additive increases to 2 wt. % in the iso-MTHPA hardener, there is an observed increase in the initial visible viscosity of the solution up to 16 s. Once the boron polymer content reaches 2 wt. %, the dilution curve becomes parallel to the x-axis, and there is no further increase in viscosity in the region of 2–5 wt. %. This phenomenon occurs due to the dissolution process of the boron polymer in the anhydride under experimental conditions, involving a swelling stage until a dynamic equilibrium is established.

The optimal dissolution conditions were determined for a solution consisting of 98.0 mass. % iso-MTHPA and 2.0 mass. % boron polymer within a temperature range of 10 to 100 °C. Figure 8 illustrates the temperature-dependent dissolution time of the boron polymer in iso-MTHPA. The analysis of the dependency reveals that once the solution is reached, the dissolution time starts to decrease within the temperature range of 40–50 °C. The resulting dilution dependency of the modifying additive boron polymer in iso-MTHPA follows a power law, with a correlation coefficient of 0.99. Considering that the decomposition processes of the iso-MTHPA hardener begin at a temperature of 55–60 °C, the optimal temperature for dilution is determined to be 55 ± 2 °C.

The improvement in mechanical properties of the epoxy–anhydride binder, achieved by adding a boron polymer, can be attributed to the chemical interaction between the boron polymer and the epoxy resin, leading to the formation of an insoluble gel fraction. The authors [15,16,17,18] conducted a study on the kinetics of curing the sol–gel system, which enabled them to establish the mechanism of interaction between the epoxy–anhydride binder and the boron polymer additive. To determine the curing mechanism, the researchers compared the IR spectra of pure polymers with the spectra of gel fractions obtained after curing with the resin.

It was found that all the polyesters and polymethylenesters of phenols and boric acid examined interacted with the epoxy resin, forming a C-C bond between the ortho- and ortho-para-positions of the phenyl radical (ortho-ortho position for the resorcinol ring) and the carbon of the epoxy group. This interaction was confirmed by changes observed in the aromatic region (675…890 cm^−1^) of the IR spectra of the gel fractions of the interaction products. Additionally, a new band appeared, characteristic of the C-O bond in the opened epoxy ring (1180 cm^−1^).

Upon analyzing the obtained results, it can be concluded that the dissolution of the modifying additive, boron polymer, in iso-MTHPA occurs at a temperature of 55 ± 2 °C over a duration of 20 h. This temperature–time condition represents the optimal condition for the dissolution of the boron polymer into iso-MTHPA. Figure 2 illustrates that the greatest increase in initial conditional viscosity is observed when the content of boron polymer in iso-MTHPA is up to 2%. The saturation of the solution and the maximum solubility of the boron polymer + iso-MTHPA mixture are defined as 2.0% by mass. Therefore, the critical point for the refrigerant additive in iso-MTHPA is 2% by mass.

### 3.2. Results of the Study on the Influence of the Modifying Additive Boron on the Technological Properties of the Epoxy–Anhydride Binder

It is known that the main requirements for epoxy binders and compounds, along with their high physico-mechanical characteristics, are their technological properties such as initial conditional viscosity, time of gel formation and time of “life” of the binder. The resulting technological parameters of the epoxy–anhydride binder with the addition of the modifying additive boron polymer are presented in Table 5.

According to Table 4, the addition of a modifying additive boron polymer in quantities from 0.50 to 1.00 mass. % does not result in a noticeable increase in the considered process parameters of the epoxy binder. The three-component epoxy binders with the modifying additive boron polymer have almost identical values for the gelatinization time and the “life” of the binder, i.e., the modifying additive boron polymer has no effect on the technological properties of the epoxy binder.

### 3.3. Results of a Study on the Influence of the Modifying Additive Boron on the Elastic-Resistant Properties of the Epoxy–Anhydride Binder

In order to assess the modifying effect of the borpolymer additive on the epoxy binder, a curing composition of the epoxy binder with and without the addition of the modifying additive borpolymer was produced. Assuming that the solubility limit of the modifying additive boron polymer iso-MTHPA is 2.00 mass. % of 100% of the mass of iso-MTHPA, the content of the modifying additive borpolymer in the epoxy composition was up to 1.00 mass. % of the mass of the epoxy binder according to Table 6.

Figure 9 and Figure 10 present the results of a study of the influence of the modifying additive boron polymer on the variation of the tensile strength and the modulus of elasticity in the transverse bending of the cured epoxy–anhydride binder.

Based on the dependence of the tensile strength at transverse bending on the content of the modifying additive boron in the epoxy binder, it follows that a maximum increase in the strength limit from 150 MPa to 181 MPa is observed for samples of epoxy binder with a content of 0.50 mass. % of boron polymer additive. Later addition of the modifying additive to the boron polymer exceeding 0.50 mass. % decreases the strength limit.

Figure 10 shows that the fracture level of the curve characterizing the dependence of the modulus of elasticity at the site of transverse bending on the content of the modifying additive in the epoxy binder coincides with the fracture level of the tensile strength at the site of transverse bending. The maximum increase in the value of the modulus of elasticity from 2450 MPa (for a pure epoxy binder without the addition of a modifying additive of boron polymer) to 3200 MPa is observed for an epoxy binder with a modifying additive content of borpolymer in the amount of 0.50 mass. %. With refrigerant additive content of 0.75 mass. % and 1.00 mass. %, the module of the modified epoxy binder fails. For an epoxy binder with a modifying additive boron polymer of 1.00 mass. %, the modulus of elasticity decreases to 1490 MPa.

Transverse bending tests on modified samples of the epoxy binder with the addition of a boron polymer show a tendency towards more brittle fracturing. When comparing modified and unmodified samples of the epoxy binder, it is worth noting the transition from a pseudoplastic to a pseudoprotic state due to the interaction of the modifying additive borpolymer with epoxy resin ED-22.

Figure 11 presents the results of experimental studies on the influence of the content of the modifying additive boron polymer on the variation of the tensile strength of the epoxy–anhydride binder.

The results of experimental tensile studies of samples of pure binder and binder with the addition of a modifying additive boron polymer showed that the maximum increase in tensile strength from 80 MPa (for samples of pure epoxy binder) to 98 MPa is observed for an epoxy binder with a modifying additive of boron polymer in the amount 0.50 mass. %. A further increase in the content of the modifying additive borpolymer results in a significant reduction in tensile strength. In tests of samples of the epoxy binder, the initial expansion of the crack is characterized by plastic deformation of the samples. The plastic deformation of the binder samples is observed until critical conditions are reached, after which a brittle break occurs. Most of the epoxy binder samples were destroyed in the area of the smallest cross section.

Experimental results of samples of pure epoxy binder and a binder with modifying additive boron polymer on impact (Charpy) are shown below in Figure 12 (GOST 4647-80). Figure 12 shows that a maximum increase in impact viscosity is observed for samples of epoxy binder with a modifying additive content of 0.50 mass. % borpolymer percent and 5.1 kJ/m^2^. Analysis of impact viscosity studies shows that the content of the modifying additive boron polymer allows us to increase this parameter by 10–15%. From the presented studies, it can be seen that the positive effect of the modification comes with a certain content of the modifying additive boron polymer, and a further increase thereof results in a reduction of the strength complex. Based on the data obtained, the introduction of the modifying additive boron polymer results in a change in the fracture and strength characteristics of the modified samples of the epoxy binders.

The efficiency of modification of composite polymer materials was evaluated by the engineering parameter of modification efficiency. The engineering parameter of the efficiency of modification Â is the ratio of the strength of the modified polymer sample to the strength of the sample without the addition of a modifier. If the engineering parameter Â is greater than 1, the modification is considered effective. The results of the evaluation of the efficiency of the modification of the epoxy binder with the boron polymer additive are presented in Table 7.

The evaluation data of the modification of the epoxy binder by engineering parameters, presented in Table 7, show that it contains a modifying additive boron polymer content of 0.25 mass. % and 0.75 mass. %, there is a slight increase in the complex of physico-mechanical properties (10%), as well as at a content of the modifying additive borpolymer in the colliery of 1.00 mass. There is a negative effect of modification of the epoxy binder. The best results (an increase from 20% to 45%) of the modification of the epoxy binder can be seen in the content of the modifying additive of boron polymer in the amount of 0.50 mass. %.

From the experimental data obtained, it can be concluded that the modifying additive boron polymer has a positive modifying effect on the epoxy binder and can be used as a modifier for epoxy binders. The results of the research show that the proposed method of modifying the epoxy binder by introducing a modifying additive boron polymer to increase the strength of the binder is promising and effective in the field of creating new composite materials. The total range of data obtained makes it possible to conclude that the best samples are samples of a binding agent to the content of the modifying additive in the amount of 0.50 mass. % of the binder mass

## 4. Discussion

Electron microscopic studies have also shown that the introduction of the boron polymer to the epoxy–anhydride binding additive alters the above molar structure of the boron polymer and causes structural formation and the formation of ordered structures. (Figure 13).

The changes in the composition structure are also evidenced by the resultant IR spectra of cured samples of epoxy binders with the addition of a modifying additive borpolymer. The IR spectrum of the pure epoxy binder is depicted in Figure 14. IR spectra with a modifying additive of 0.50 mass. %, 0.75 mass. % and 1.00 mass. % are shown in Figure 15, Figure 16 and Figure 17. The evaluation of the change in the structure of the polymer on the IR spectra obtained was evaluated by the change in the intensity of the bands in the range considered.

From the IR spectra data (Figure 14, Figure 15, Figure 16 and Figure 17) it can be seen that when adding the modifying additive boron polymer to the epoxy binder, there is a decrease in the intensity of the band at 3013.3–3600.0 cm^−1^ with a vertex at 3435.38 cm^−1^ that characterizes the intermolecular hydrogen bond of the epoxy binder. That is, in the process of curing the epoxy binder, the modifying additive borpolymer reacts chemically with epoxy resin and thus contributes to the formation of a new three-dimensional mesh structure of the composition.

## 5. Conclusions

The research carried out on the influence of the modification of the epoxy–anhydride binder for the addition of borpolymer leads to the following conclusions.

The results of the experimental studies showed that the proposed method of modifying the epoxy binder by introducing a modifying additive boron polymer to enhance strength was effective.

Experimental studies on the dissolution process of the modifying additive borpolymer in the components of the epoxy–anhydride binder have shown that the process of dissolution of the borpolymer only occurs in the iso-MTHPA curing agent. The optimal temperature–time conditions for solubility of the borpolymer in the iso-MTHPA, providing complete solubility of the modifying additive of the boron polymer in iso-MTHPA, is dissolution at a temperature of 55 ± 2 °C for 20 h.Experimental studies of viscosity, the time of “life” and the time of gel formation of the modified epoxy binder show that modification of the epoxy binder with the borpolymer additive is carried out without changing the technological parameters of the original epoxy composition.Based on the obtained experimental physico-mechanical studies of modified epoxy compositions, it follows that the optimal content of the modifying additive borpolymer in the composition of the epoxy binder is 0.50 mass. %. With a modifying additive content of 0.50 mass. % in the composition of the epoxy binder, the tensile strength at transverse bending is increased to 190 MPa, the modulus of elasticity to 3200 MPa, the tensile strength to 0.8 MPa and the impact viscosity (Charpy) to 5.1 kJ/m^2^.Considering all the above, it follows that the increased strength of the epoxy binders when a modifying additive is introduced into them is due to the possible formation of additional transverse stitches between the carbon epoxy group and carbon, in a stable position in relation to the hydroxyl group of phenol when the composition is cured. Indirect evidence of this is the change in the physico-mechanical performance of the modified samples, in particular the increase in the strength and the elasticity module at the transverse bending site by 15 percent, which destroys 20 percent of tensile loads, and impact (Charpy) by 10%. Infrared spectra indicate the chemical interaction of the modifying additive boron polymer with the binder.

## Figures and Tables

**Figure 1 polymers-15-02632-f001:**
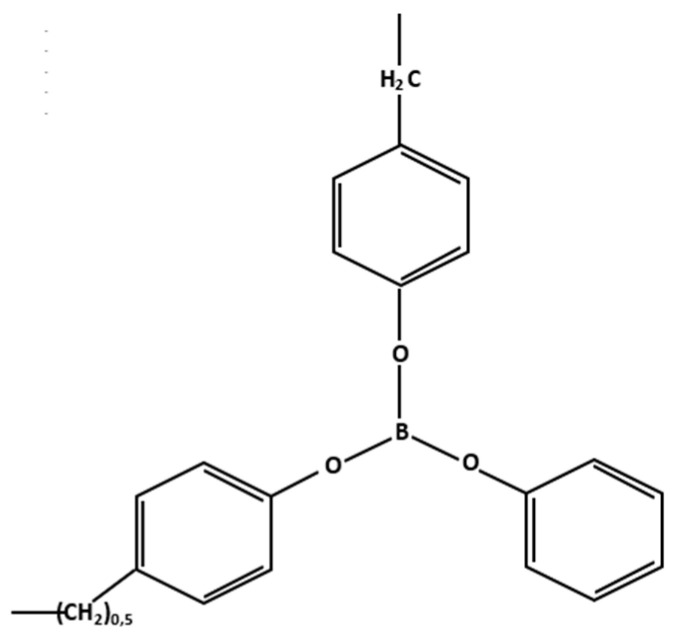
Structural formula of the modifying additive borpolymer.

**Figure 2 polymers-15-02632-f002:**
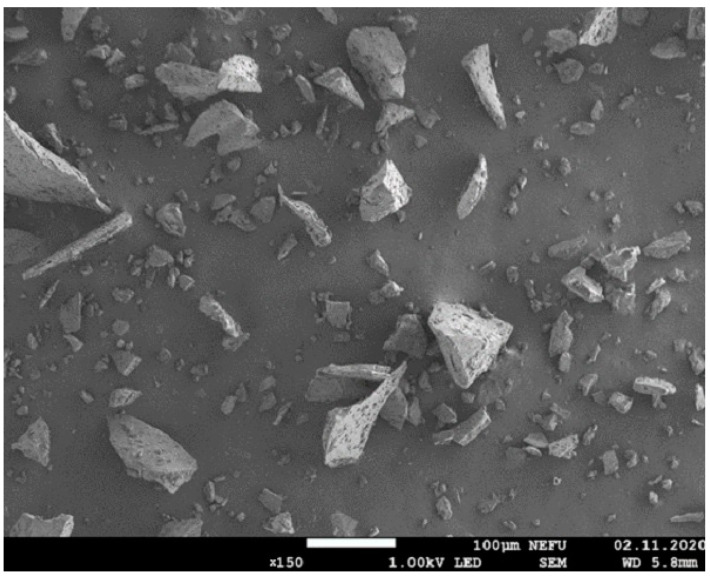
Borpolymer powder with 150 times magnification.

**Figure 3 polymers-15-02632-f003:**
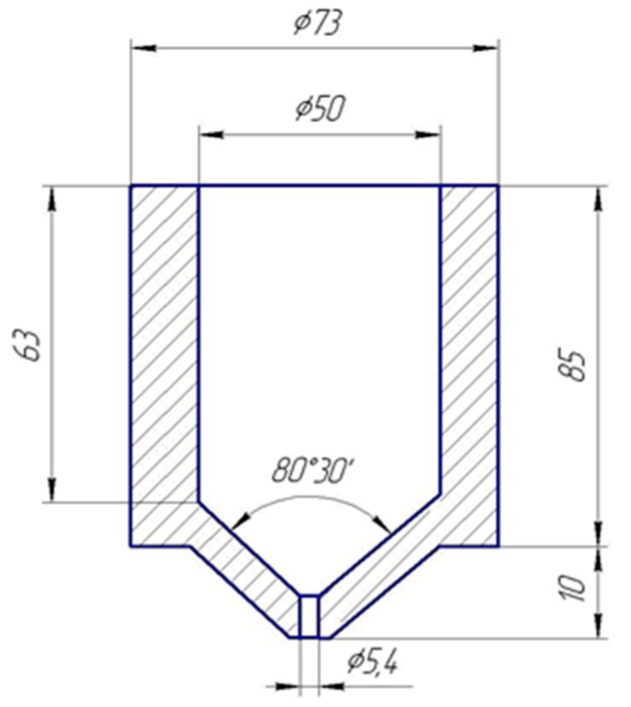
Schematic viscometer VZ-1.

**Figure 4 polymers-15-02632-f004:**
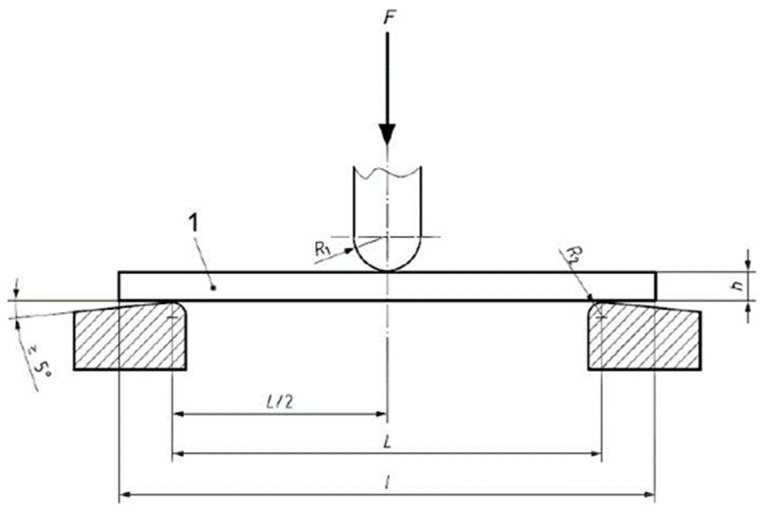
Static bending test.

**Figure 5 polymers-15-02632-f005:**
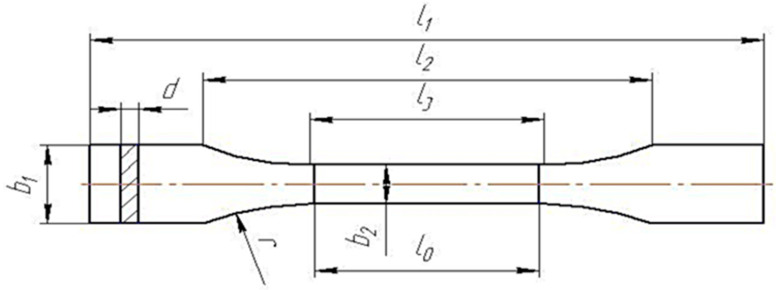
Tensile test sample GOST 11262-80: *l_0_
*= 50 ± 0.5 mm, *l_1_
*= 150 ± 0.5 mm, *l*_2_ = 115 ± 0.5 mm, *l*_3_ = 60 ± 0.5 mm, *b*_2_ = 10 ± 0.5 mm; *b*_1_ = 20 ± 0.5 mm, *d* = 4 ± 0.4 mm, *r* = 60 mm.

**Figure 6 polymers-15-02632-f006:**
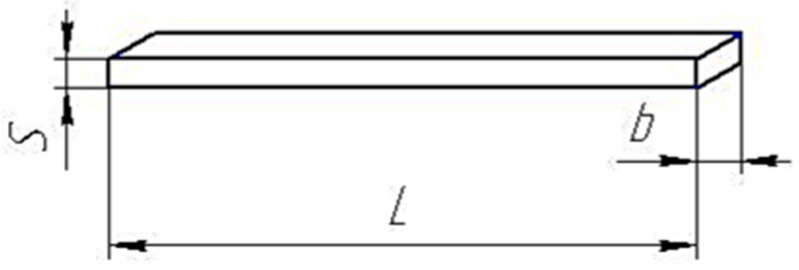
Cut-free sample to determine the Charpy impact viscosity at GOST 4647-80, *L* = 80 ± 0.5 mm; *b* = 10 ± 0.5 mm; *S* = 4 ± 0.2 mm.

**Figure 7 polymers-15-02632-f007:**
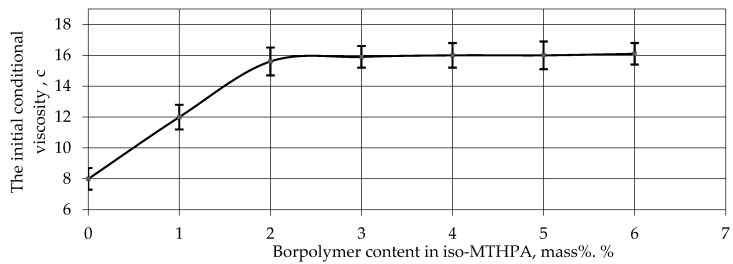
The dependence of the iso-MTHPA isopolymer content on the initial conditional viscosity of the mixture at 40 °C.

**Figure 8 polymers-15-02632-f008:**
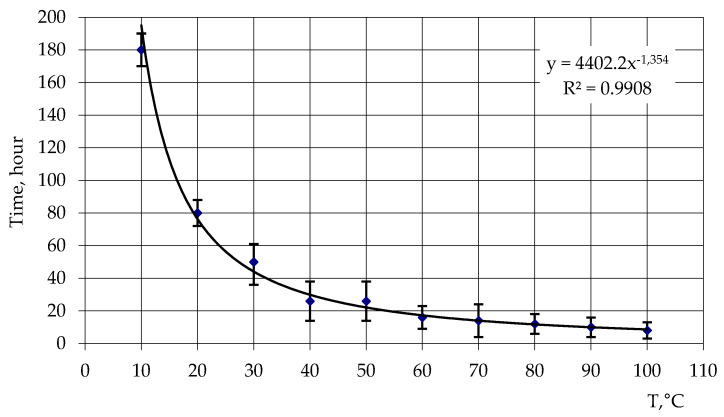
Dilution time of solution of 98 mass. %. iso-MTHPA and 2 mass. % borpolymer, from temperature.

**Figure 9 polymers-15-02632-f009:**
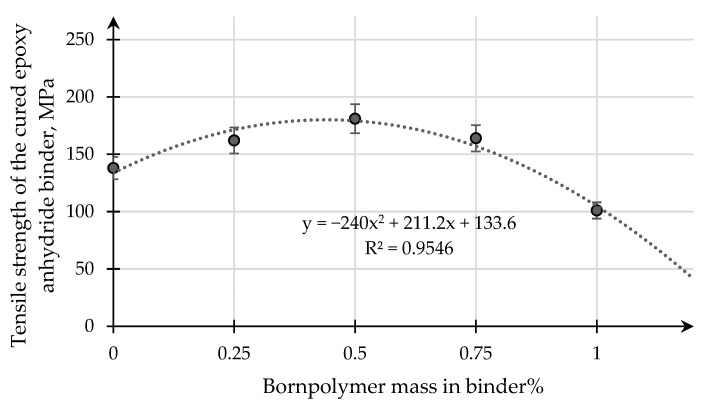
Transverse bending strength depends on the content of the modifying additive in the epoxy–anhydride binder. Polynomial distribution (trend line) value of approximation confidence 0.95.

**Figure 10 polymers-15-02632-f010:**
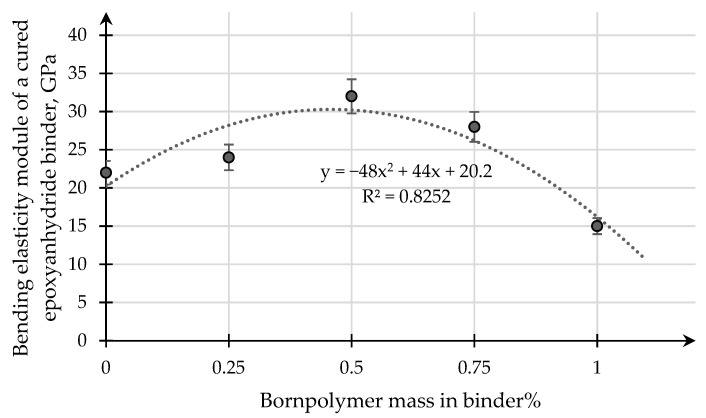
The dependence of the elastic modulus at transverse bending on the content of the modifying additive in the epoxy–anhydride binder.

**Figure 11 polymers-15-02632-f011:**
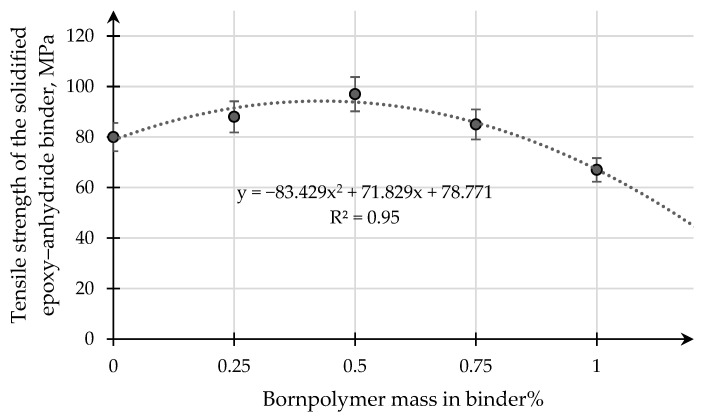
Tensile strength dependent on the content of the modifying additive in the epoxy–anhydride binder.

**Figure 12 polymers-15-02632-f012:**
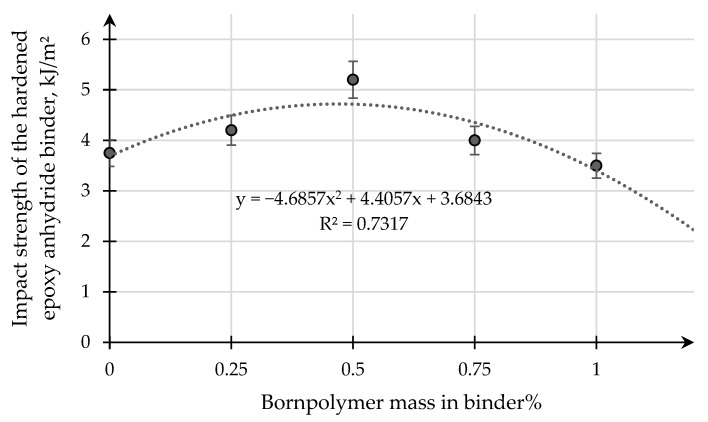
Impact viscosity (Charpy) depends on the concentration of the modifying additive boron polymer in the epoxy binder.

**Figure 13 polymers-15-02632-f013:**
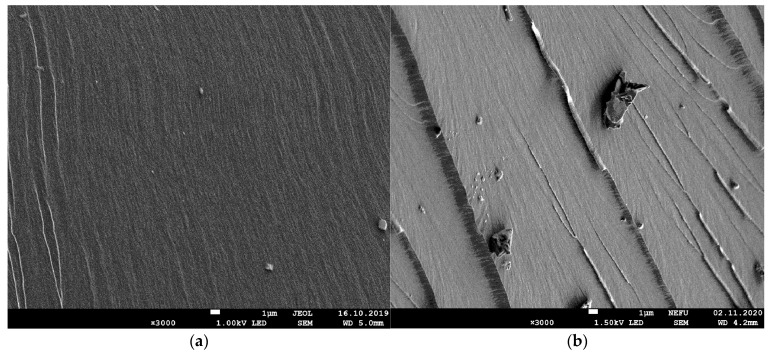
The microstructure of the degradation surface in the original binder (**a**), in the modified combo polymer 0.5% by mass, and (**b**) with an increase of 3000 times.

**Figure 14 polymers-15-02632-f014:**
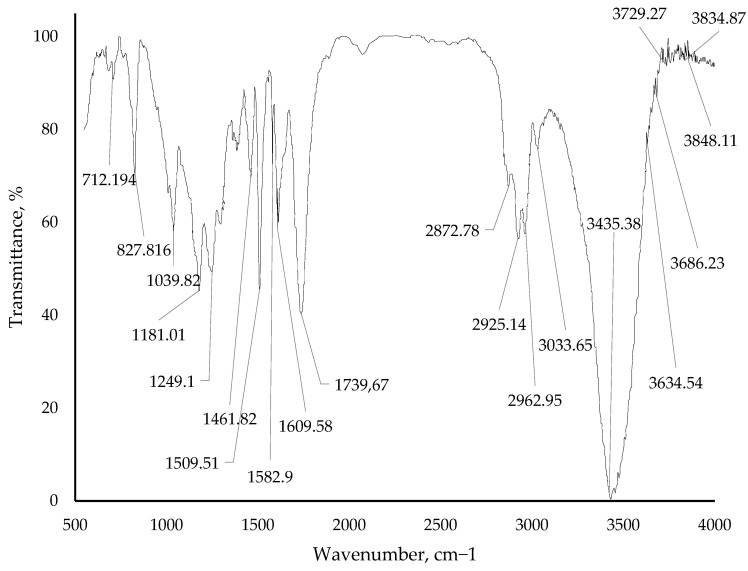
IR spectrum of epoxy binder without adding modifying additive boron polymer.

**Figure 15 polymers-15-02632-f015:**
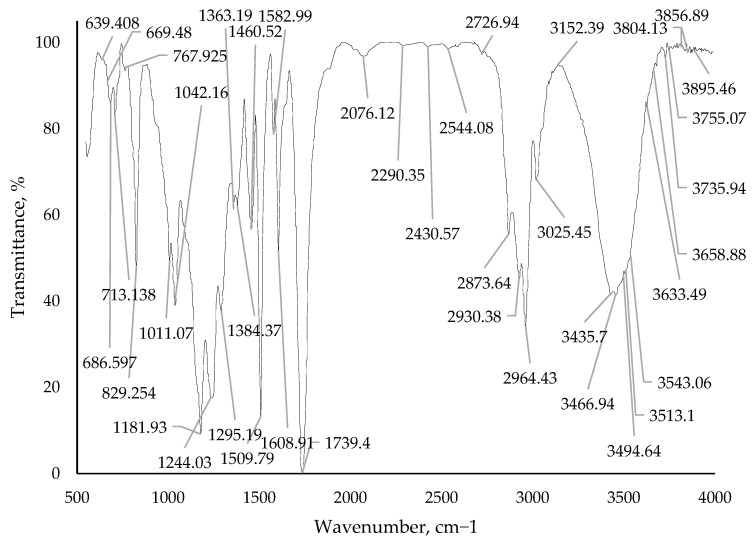
IR spectrum of epoxy binder with addition of modifying additive boron polymer in a quantity of 0.50 mass. %.

**Figure 16 polymers-15-02632-f016:**
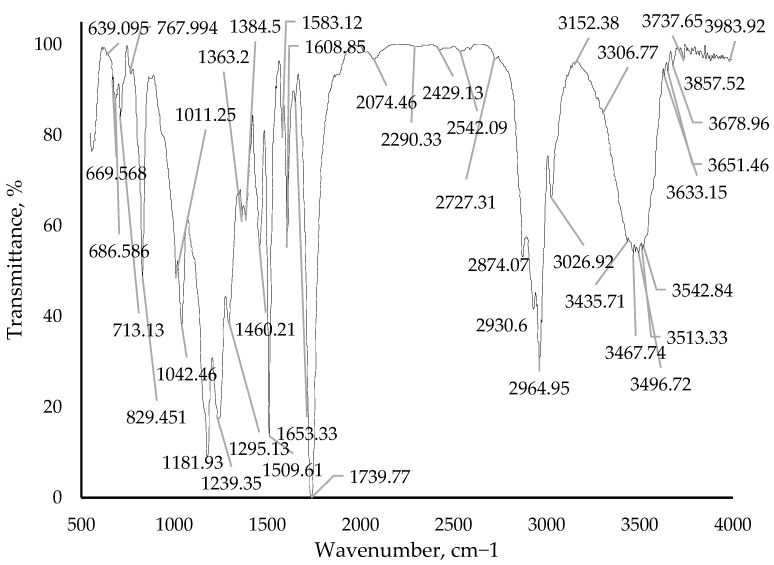
IR spectrum of epoxy binder with addition of modifying additive boron polymer in an amount of 0.75 mass. %.

**Figure 17 polymers-15-02632-f017:**
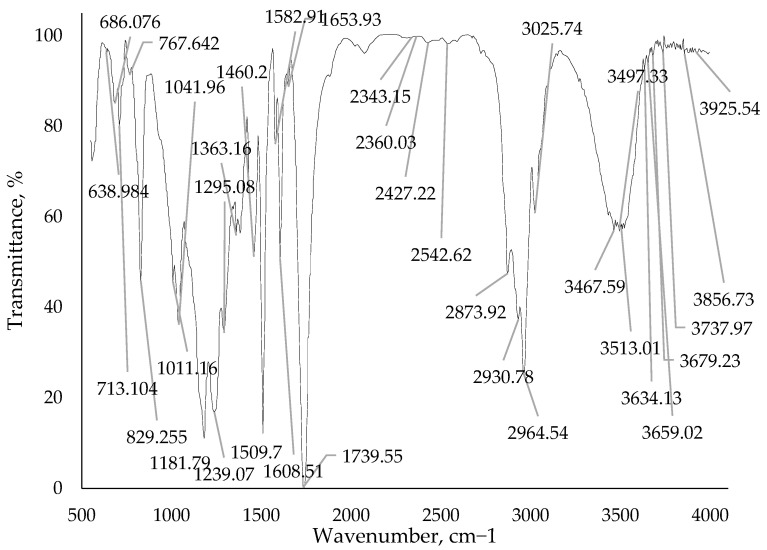
IR spectrum of epoxy binder with addition of modifying additive boron polymer in a quantity of 1.00 mass. %.

**Table 1 polymers-15-02632-t001:** Characteristics of binder components [19,20,21].

Component	Density [kg/m^3^]	Dynamic Viscosity (25 °C), [Pa∙s]	Gelatinization Time, [h]
Epoxy resin ED-22	1165	1170–1230	970
iso-MTHPA	7–12	0.12	0.35–0.65
Agidol-53	16.0 (100 °C)	5.5–8.0 (150 °C)	0.6–1.3 (80 °C)

**Table 2 polymers-15-02632-t002:** Physico-chemical properties of the modifying additive boron polymer.

Indicator Name	Average Molecular Weight	Melting Point	Solubility
Value	2500–3000	150–160	Soluble in polar organic solvents, weakly soluble in non-polar organic

**Table 3 polymers-15-02632-t003:** Test program for the determination of the impact of the boron polymer on manufacturability and elastic strength properties of resin.

Parameter	Method of Measuring	Note
time of “life”	Paint and Lacquer Materials. Method for determination of relative viscosity GOST 8420-74 (ISO 2431-84)	The maximum viscosity value is taken as the time of “life” of polymer resin, at which the composition becomes unsuitable for further processing
time of gel formation	to test the time of gel formation they use a device heating some volume of polymer binder, which extends not more than 20 mm above the surface of the binder.	During the gel-forming period, a time interval is taken from the time when the binder is injected into the container, when the binder removed from the receptacle is terminated.
primary relative viscosity	Paint and Lacquer Materials. Method for determination of relative viscosity	The conditional viscosity of free-flowing materials is the time of continuous expiration in seconds of a certain volume of test material through a calibrated nozzle of viscometer.
bending strength	Plastics. Method of static bending test GOST 4648-2014 (ISO 178:2010)	
Module of elasticity and ultimate tensile strength	Plastics. Tensile test method GOST 11262-2017 (ISO 527-2:2012)	
Impact strength of samples without incisions	Plastics. Method for determination of Charpy’s impact strength GOST 4647-80	

**Table 4 polymers-15-02632-t004:** Initial conditional viscosity values, defined at 40 °C, for boron polymer + ED22 and boron polymer + Agidol-53 solutions, which have been aged at 50 ± 2 °C for 20 days.

Indicator	Value of the Indicator				
Boron polymer content in ED-22/Agidol-53, mass. %	1.0	2.0	3.0	4.0	5.0
Initial conditional viscosity of boron polymer + ED-22 solution, s	230 ± 5	229 ± 3	231 ± 5	228 ± 4	230 ± 5
Initial conditional viscosity of borpolymer+ Agidol-53 solution, s	12.1 ± 1	12 ± 0.5	12 ± 0.4	12 ± 0.3	12 ± 1

**Table 5 polymers-15-02632-t005:** Technological parameters of epoxy–anhydride binder with addition of modifying additive of boron polymer.

Borpolymer Content in Epoxyanhydride Binder, Mass. %	Initial Conditional Viscosity, c	Time of Gel-Formation, c	Time of “Life”, c
0	56 ± 3	360 ± 5	≈25,200
0.5	84 ± 2	382 ± 4	≈25,200
0.75	98 ± 3	375 ± 7	≈25,200
1.0	102 ± 4	380 ± 8	≈25,200

**Table 6 polymers-15-02632-t006:** Compositions of epoxy–anhydride binders.

Composition N°	Components, Mass. %			
	ED–22	Iso-MTHPA	Agidole-53	Borpolmer
	56.7 ± 0.2	42.5 ± 0.2	0.8 ± 0.1	0
1	56.56 ± 0.2	42.40 ± 0.2	0.79 ± 0.1	0.25 ± 0.1
2	56.42 ± 0.2	42.29 ± 0.2	0.79 ± 0.1	0.5 ± 0.1
3	56.28 ± 0.2	42.18 ± 0.2	0.79 ± 0.1	0.75 ± 0.1
4	56.10 ± 0.2	42.10 ± 0.2	0.79 ± 0.1	1.0 ± 0.1

**Table 7 polymers-15-02632-t007:** Evaluation of the engineering efficiency of modification of the epoxy binder with the borpolymer additive.

Test Method	Value Â for Modifying Additive Content of Epoxy Binder in Quantity		
	0.25 Mass. %	0.50 Mass. %	0.75 Mass. %
Tensile strength at transverse bending	1.17	1.37	1.24
Transverse bend elastic module	1.14	1.45	1.32
Tensile strength	1.11	1.22	1.05
Impact strength (Charpy)	1.13	1.34	1.05

## Data Availability

The data presented in this study are available on request from the corresponding author.

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
