# Peer review of "Reinforced Epoxy Binder Modified with Borpolymer"

_polymers, 2023, doi:10.3390/polym15122632_

Round 1

Reviewer 1 Report (Previous Reviewer 2)

We encountered several typos (e.g., "epoxy- anhydride resins have o lot of drawbacks" in line 57). We highly advise that the authors perform meticulous language editing prior to publication.   

We encountered several typos (e.g., "epoxy- anhydride resins have o lot of drawbacks" in line 57). We highly advise that the authors perform meticulous language editing prior to publication.   

Author Response

Dear reviewer, we would like to express our gratitude to you for the re-review of the manuscript. We are providing you with the corrected manuscript

Reviewer 2 Report (Previous Reviewer 3)

The authors have provided a revised manuscript, but the reviewer did not see a point-to-point response. The following response have not been well reflected in the previous comments.

1. The introduction section has not been substantially enriched and improved, and the current content is unable to provide readers with a comprehensive information. Please enrich and summarize this paper in detail based on the previous comments.

2. For the part 3.1 of the results and discussion, it should be written paragraph by paragraph. At the same time, it should be noted that more paragraphs only including few sentences are not desirable.

3. In figure 7, please explain why when the content exceeds 2%, the initial conditional viscosity did not change.

4. For figure 8-11, please provide detailed and convincing mechanism explanation of mechanical property evolution with the bornpolymer mass in binder, including the physical and chemical combination mechanism between epoxy resin and additives.

It needs the minor improvement. 

Author Response

Dear reviewer, we are providing you with the corrected manuscript and answers to the following questions.

  1. The introduction section has not been substantially enriched and improved, and the current content is unable to provide readers with a comprehensive information. Please enrich and summarize this paper in detail based on the previous comments.

Answer: The introduction section has been updated, including information on the characteristics, advantages, and applications of epoxy resin. The methods of epoxy resin modification, including chemical and physical modifications, have been described. (Changes are highlighted with yellow markers).

1. Introduction

World practice in the production of structural composite materials shows that the most accessible and well-recommended binders are currently epoxy binders and heat curing compounds [1,2,9]. Their widespread use in engineering is associated, firstly, with the high manufacturability of epoxy resins, and secondly, with a unique combination of performance characteristics of products for their curing. Epoxy resins are oligomers that contain epoxy groups and are capable of forming crosslinked polymers under the influence of curing agents (such as anhydrides, amines, polyamines, etc.). Epoxy resins based on the polycondensation of epichlorohydrin with phenols, typically bisphenol A, have gained the most widespread use. The high reactivity of the epoxy group and the thermodynamic compatibility of epoxy oligomers with many substances make it possible to use a variety of hardeners and carry out curing reactions under various technological conditions. Such features of the synthesis processes as the absence of volatile products and low shrinkage are equally important.

Epoxy polymers have high values of static and impact strength, hardness and wear resistance. They are characterized by significant temperature and heat resistance. Many hard surfaces form strong adhesive bonds with epoxy polymers, which explains their use as adhesives, paints, coatings, and binders in composites.

In the manufacture of fiberglass plastics, in particular, 3-components epoxy resin with dicarbonic acid anhydrides as a hardening agent is widely used. These epoxy- anhydride resins obtain high strength properties, low settling, high adhesion to fibrous fillers. However, epoxy- anhydride resins have some drawbacks, e.g low heat resistance (within 120ºС), resistance to aggressive environments, flammability etc.

Further extension of the field of application of epoxy- anhydride resins is possible by means of chemical and physico-chemical modification, changing properties of the epoxy binder [3-8]. Chemical modification involves altering the polymer network structure by adding compounds that become part of the polymer composition. For example, the addition of laproxides (simple polyether alcohols containing glycidyl groups, such as glycerin anhydride) provides elasticity to the cured resin by increasing the molecular weight of the interstitial fragment but reduces its water resistance. Adding halogen and phosphorus organic compounds increases the flame retardancy of the resin. Incorporating phenol-formaldehyde resins allows the epoxy resin to cure through direct heating without a curing agent, providing greater stiffness, improved anti-friction properties, but reduces impact viscosity. Physical modification is achieved by adding substances to the resin that do not form a chemical bond with the binder. For example, adding rubber increases the impact viscosity of the cured resin. The addition of colloidal titanium dioxide increases its refractive index and imparts opacity to ultraviolet radiation. Search for chemical substances, enabled to modify the structure and correspondingly properties of a resin, is an important research and implementation objective.

It is known that boron compounds [13], in particular, polyesters and boric acid polymethylene esters are well adhesive to glass, metals, wood and are now mainly used as adhesive compositions, plasticizers, lubricants, emulsifiers etc. Interesting and unique properties has the boron compound synthesized by authors [14-20] in their work: boric acid polymethylene-p-triphenyl ether (boron compound).

Thus, the relevant research objective is to assess the effectiveness of the application of boric acid polymethylene-p-triphenyl ether for modification of structure and properties of epoxy- anhydride resins and development of a combination technology through the solubility phase of boric acid polymethylene-p-triphenyl ether additives in components.

The innovativeness of the proposed solution lies in the use of a new synthesized compound, which is not currently used by any research group involved in the modification of epoxy binders.

  1. For the part 3.1 of the results and discussion, it should be written paragraph by paragraph. At the same time, it should be noted that more paragraphs only including few sentences are not desirable.

Answer: Section 3.1 has been updated and corrected according to the comments. (Changes are highlighted with yellow markers).

  1. In figure 7, please explain why when the content exceeds 2%, the initial conditional viscosity did not change.

Answer: In section 3.1, under Figure 7, the justification for the absence of initial conditional viscosity change at a concentration above 2% has been described. (Changes are highlighted with yellow markers).

Figure 7 illustrates the change in initial conditional viscosity of the boron polymer + iso-MTGFA solution. Under the experimental conditions, the maximum boron polymer filling degree was 5 wt. %. The figure indicates that the pure iso-MTHPA hardener has an initial conditional viscosity of 8 seconds. As the content of the modifying additive increases to 2 wt. % in the iso-MTHPA hardener, there is an observed increase in the initial visible viscosity of the solution up to 16 seconds. Once the boron polymer content reaches 2 wt. %, the dilution curve becomes parallel to the x-axis, and there is no further increase in viscosity in the region of 2-5 wt. %. This phenomenon occurs due to the dissolution process of the boron polymer in the anhydride under experimental conditions, involving a swelling stage until dynamic equilibrium is established.

The optimal dissolution conditions were determined for a solution consisting of 98.0 mass. % iso-MTHPA and 2.0 mass. % boron polymer within a temperature range of 10 to 100°C. Figure 8 illustrates the temperature-dependent dissolution time of the boron polymer in iso-MTHPA. The analysis of the dependency reveals that once the solution is reached, the dissolution time starts to decrease within the temperature range of 40-50°C. The resulting dilution dependency of the boron polymer modifying additive in iso-MTHPA follows a power law, with a correlation coefficient of 0.99. Considering that the decomposition processes of the iso-MTHPA hardener begin at a temperature of 55-60°C, the optimal temperature for dilution is determined to be 55±2 °C.

  1. For figure 8-11, please provide detailed and convincing mechanism explanation of mechanical property evolution with the bornpolymer mass in binder, including the physical and chemical combination mechanism between epoxy resin and additives.

Answer: The mechanism of interaction, both from a physical and chemical standpoint, has been established by our colleagues and is not the focus of our research. Therefore, we acknowledge the observed changes in mechanical properties and assess the effectiveness of the additive.

The improvement in the mechanical properties of the epoxy anhydride binder, resulting from the addition of boropolymer, can be attributed to the chemical interaction between the boropolymer and the epoxy resin, leading to the formation of an insoluble gel fraction. Previous studies conducted by the authors [18-21] on the kinetics of the curing process using the sol-gel method have provided insights into the interaction mechanism between the epoxy anhydride binder and the boropolymer additive. To investigate the curing mechanism, the researchers compared the infrared (IR) spectra of pure polymers with the spectra of gel fractions obtained after curing with the resin.

The findings indicate that all the studied polyesters and polymethylenesters derived from phenols and boric acid interact with the epoxy resin, resulting in the formation of a C-C bond between the o- and o,p-positions of the phenyl radical (o,o-position for the resorcinol ring) and the carbon atom of the epoxy group. These interactions are confirmed by changes in the aromatic region (675...890 cm^-1) of the IR spectra of the gel fractions of the interaction products, as well as the appearance of a new band characteristic of the C-O bond in the opened epoxy cycle (1180 cm^-1).

In summary, the boropolymer interacts with the epoxy resin, forming a C-C bond between the o- and o,p-positions of the phenyl radical (o,o-position for the resorcinol ring) and the carbon atom of the epoxy group. This interaction is supported by changes in the aromatic region of the IR spectra of the gel fractions of the interaction products, as well as the presence of a new band characteristic of the C-O bond in the opened epoxy cycle. (Changes are highlighted with yellow markers).

The improvement in mechanical properties of the epoxy anhydride binder, achieved by adding boron polymer, can be attributed to the chemical interaction between the boron polymer and epoxy resin, leading to the formation of an insoluble gel fraction. The authors [18-21] conducted a study on the kinetics of curing the sol-gel system, which enabled them to establish the mechanism of interaction between the epoxy anhydride binder and the boron polymeradditive. To determine the curing mechanism, the researchers compared the IR spectra of pure polymers with the spectra of gel fractions obtained after curing with the resin.

It was found that all the polyesters and polymethylenesters of phenols and boric acid examined interacted with the epoxy resin, forming a C-C bond between the ortho- and ortho-para-positions of the phenyl radical (ortho-ortho position for the resorcinol ring) and the carbon of the epoxy group. This interaction was confirmed by changes observed in the aromatic region (675...890 cm^-1) of the IR spectra of the gel fractions of the interaction products. Additionally, a new band appeared, characteristic of the C-O bond in the opened epoxy ring (1180 cm^-1).

Upon analyzing the obtained results, it can be concluded that the dissolution of the modifying additive, boron polymer, in iso-MTHPA occurs at a temperature of 55±2 °C over a duration of 20 hours. This temperature-time condition represents the optimal condition for the dissolution of the boron polymer into iso-MTHPA. Figure 2 illustrates that the greatest increase in initial conditional viscosity is observed when the content of boron polymer in iso-MTHPA is up to 2%. The saturation of the solution and the maximum solubility of the boron polymer + iso-MTHPA mixture are defined as 2.0% by mass. Therefore, the critical point for the refrigerant additive in iso-MTHPA is 2% by mass.

Round 2

Reviewer 2 Report (Previous Reviewer 3)

Accepted.

ok

This manuscript is a resubmission of an earlier submission. The following is a list of the peer review reports and author responses from that submission.

Round 1

Reviewer 1 Report

The manuscript is very poorly written and hard to understand. The abstract doesn't highlight the main objective and findings of this work. The title of the manuscript doenst make sense. the topic of research seems to be not very well studied as the authors have only 10 references, and few of them are self-citations and many from obscure references.

Author Response

Good day! Thank you for your up to date comments. The title of the article has been updated, the introduction reflecting the essence of the study. Updated references to literature. Your corrections are highlighted in yellow.

Reviewer 2 Report

After carefully reviewing the manuscript, I cannot recommend this manuscript to be further considered for publication in Polymers.  

1.      Title doesn’t properly describe the work done, and its grammar need to be checked, and should be reconsidered.

2.      Abstract doesn't highlight the key findings in this work.

3.      The introduction is under cited and is very short, and does not provide important information regarding the background and reasoning for why this work is being reported in the first place (with the exception of mentioning epoxy –anhydride resins).

4.      Figure 2 (SEM micrographs) seem to be manipulated (contrast etc.). If so, please note what done to the image.

5.      The use of words as "Technological parameters", "technological properties" and so on is very obscure and is used in the manuscript to describe various phenomena. 

6.      There are only 10 citations, most of them are self-citations, and are out dated (except for 2).

Author Response

Good day! Thank you for your timely comments on the manuscript. Please read the revised manuscript with the updated title, annotation, introduction, bibliography. All fixes are highlighted in yellow.

Reviewer 3 Report

The influence of the modification of epoxy resin for fiberglass by a boron polymer additive was investigated. However, the current presentation of research work needs the further improvement. The authors should consider the following comments to make necessary supplements. 

1. The abstract does not convey some important information and exposes some important results or conclusions related to this article. In addition, the research background and significance should also be further added. It is suggested to consider the above comments to further enrich the writing of the abstract.

2. The introduction writing must be further enriched to make necessary supplement according to the following comments.

1) The performance, advantages and main application fields of epoxy resin should be further analyzed and summarized, so that readers can better understand the basic information about epoxy resin.

2) The performance improvement methods of epoxy resin, including physical, chemical methods and the other methods, should be analyzed and summarized in detail. The further improvement mechanism of mechanical and thermal properties should also be further clarified through the summary of other work.

3) The problems to be solved in the current main research work should be further clarified aiming at the problems that have not been solved in the literature, and the contribution and innovation of this paper should be further highlighted.

It is suggested that the authors make necessary improvement to fill this gap based on the above comments through reviewing the latest research work, such as Nanomaterials 11 (5), 1234. Progress in Materials Science, 2022: 100977. Polymers 14 (6), 1087.

3. The basic information of materials, such as materials parameters and sources used in this paper should be provided in detail in Part 2.

4. The current writing on performance testing methods is relatively confusing. It is recommended to write the test method one by one according to the test content, including the test details, reference standard, sample number, etc.

5. The secondary title in the results of Part 3 should focus on the performance research, not the results, and the authors are suggested to modify them.

6. For the part 3.1 of the results and discussion, it should be written paragraph by paragraph. At the same time, it should be noted that more paragraphs only including few sentences are not desirable.

7. In figure 7, please explain why when the content exceeds 2%, the initial conditional viscosity did not change.

8. The tables should be further standardized and professional. Generally, three-line tables are OK.

9. For figure 8-11, please provide detailed and convincing mechanism explanation of mechanical property evolution with the bornpolymer mass in binder, including the physical and chemical combination mechanism between epoxy resin and additives.

10. Please improve the image quality of Figure 13~Figure 16.

Author Response

Good day! Thank you for your timely comments on the manuscript. Please read the revised manuscript with the updated title, annotation, introduction, bibliography, figures. All fixes are highlighted in yellow.
